# COLLIE: SYSTEMATIC CONSTRUCTION OF CONSTRAINED TEXT GENERATION TASKS

**Shunyu Yao**[*]  **Howard Chen**[*]  **Austin W. Hanjie**[*]  **Runzhe Yang**[*]  **Karthik Narasimhan**
Department of Computer Science, Princeton University
`{shunyuy, hc22, hjwang, runzhey, karthikn}@princeton.edu`

## ABSTRACT

With the rapid improvement of large language models capabilities, there has been increasing interest in challenging constrained text generation problems. However, existing benchmarks for constrained generation usually focus on fixed constraint types (e.g. generate a sentence containing certain words) that have proved to be easy for state-of-the-art models like GPT-4. We present COLLIE, a grammar-based framework that allows the specification of rich, compositional constraints with diverse generation levels (word, sentence, paragraph, passage) and modeling challenges (e.g. language understanding, logical reasoning, counting, semantic planning). We also develop tools for automatic extraction of task instances given a constraint structure and a raw text corpus. Using COLLIE, we compile the COLLIE-v1 dataset with 2,080 instances comprising 13 constraint structures. We perform systematic experiments across five state-of-the-art instruction-tuned language models and analyze their performances to reveal shortcomings. COLLIE is designed to be extensible and lightweight, and we hope the community finds it useful to develop more complex constraints and evaluations in the future.

## 1 INTRODUCTION

Large language models (LLMs) are increasingly capable of generating coherent and fluent text when provided with high-level prompts (OpenAI, 2023a). Such capabilities have raised the bar for automated text generation, allowing us to explore more nuanced ways of utilizing LMs. One such line of inquiry is constrained text generation, whereby the LM is asked to adhere to a particular topic (Keskar et al., 2019; Dathathri et al., 2020), or avoid using certain words (Lu et al., 2021; 2022). However, these works scratch the surface of a broader phenomenon — LMs do not just generate text, as evidenced by their use in more structured tasks like problem solving (Yao et al., 2022), code generation (Chen et al., 2022b) and even tool use through API calls (Schick et al., 2023).

This raises a natural question — '*what is the next iteration of text generation benchmarks that can evaluate these advanced capabilities in LLMs*'? We posit that one direction is incorporating logical and compositional challenges via constrained text generation. Existing benchmarks for constrained generation, however, focus only on particular constraint types, require tailored pipelines to collect data and annotations, and/or can only evaluate a specific aspect of LM strengths (Lin et al., 2020; Chen et al., 2022a). They also suffer from challenges in scalable dataset construction.

In this paper, we propose COLLIE, a grammar-based framework that enables systematic construction of compositional constraints over diverse generation levels (e.g., words, sentences, paragraphs) and semantic requirements (e.g., language understanding, logical reasoning, counting). Operationally, COLLIE allows researchers to 1) easily specify constraint templates, and then automatically 2) extract constraint values from language corpora, 3) render them into natural language instructions, and 4) evaluate model generations against the constraint instructions.

Existing benchmarks for constrained generation focus only on particular constraint types and formats (e.g., "generate a sentence with words..."). These limitations mean that benchmarks become quickly obsolete as LLMs progress. In contrast, the modular and extensible design of COLLIE allows the

---

[*]Equal contribution. Project site with code and data: `https://collie-benchmark.github.io`.
Collie is a herding dog that can help guide domesticated animals like llamas and alpacas.

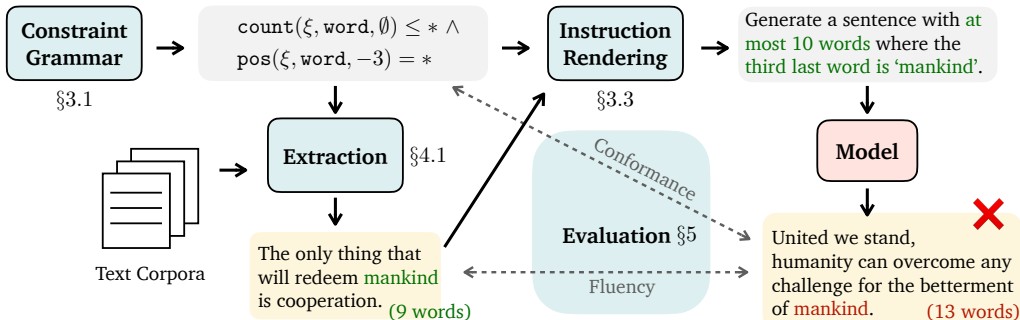

Figure 1: Our **COLLIE framework** for constraint structure specification, ground truth extraction, instruction rendering, and evaluation. First, the user specifies the constraint structure without a specific target value (expressed in ∗). Second, the constraint structure is used to extract ground truth examples from text corpora that contain the target values. Third, the constraint structure and target values are rendered into a natural language instruction. Finally, the model's generation is evaluated against the constraint and the ground truth. The model (`gpt-3.5-turbo`) violates the constraints by exceeding word limits and leaving the word 'mankind' at the end instead of the specified position.

broader NLP community to contribute additional constraints that can co-evolve with LLM capabilities over time, while also providing a convenient endpoint for users that only want to evaluate their model without developing their own constraints. The flexibility of such a grammar-based framework may not only be useful for evaluation, but also in practice (e.g. word constraints, words blacklist, etc.).

We construct the dataset COLLIE-v1 with 2,080 constraint instances across 13 different types, using three different corpora: Wikipedia (Foundation, 2022), CC-News (Hamborg et al., 2017), and Project Gutenberg (Brooke et al., 2015). We perform zero-shot evaluations of five state-of-the-art LLMs of varying sizes including GPT-4 (OpenAI, 2023a) and PaLM (Anil et al., 2023). While GPT-4 comparatively performs the best, it still achieves an average constraint satisfaction rate of only 50.9%. We find that challenges correlate with position – for instance, instructing models to begin a sentence with a specific word leads to a 100% success rate for GPT-4, while asking models to end a sentence with a particular word results in a success rate of 40%-60%. These insights can help us diagnose LLMs, which in turn can improve LLM capabilities, and further advance the benchmark itself.

To summarize, we make the following contributions: (1) We introduce COLLIE, a framework for systematic generation of compositional constraints, that is flexible and extensible. (2) We use COLLIE to curate a new dataset COLLIE-v1 comprising of 13 constraint structures. (3) We perform a comprehensive evaluation of five state-of-the-art LLMs of varying sizes and provide useful insights for both model and benchmark development in the future.

## 2 RELATED WORK

**Constrained text generation (CTG).** Early work in controllable text generation used control codes to steer the generation towards desired topics or to reduce undesirable content, by controlling for broad attributes such as sentiment or toxicity (Hu et al., 2017; Keskar et al., 2019; Dathathri et al., 2020; Krause et al., 2021). Other work on constrained decoding provides to the language model a collection of lexical items as constraints to be included or excluded in the final generated text (Hokamp and Liu, 2017; Hasler et al., 2018; Dinu et al., 2019; Hu et al., 2019; Lin et al., 2020; Lu et al., 2021; 2022; Li et al., 2022b). Recent advances in instruction tuning LLMs (Ouyang et al., 2022) have brought major improvements to controllability. These advancements have made it challenging to use existing controllable generation datasets to fully assess the capabilities of modern LLMs. InstructCTG (Zhou et al., 2023) is a concurrent work that also constructed a dataset with text constraints. However, it mainly focuses on synthesizing 5 types of simple CTG instructions for tuning small language models such as T5-11B (Raffel et al., 2020), whereas COLLIE serves to construct much more challenging and open-ended CTG tasks to evaluate and diagnose start-of-the-art LLMs like GPT-4 (OpenAI, 2023b). Lastly, there is a line of work that sets up CTG in more practical downstream applications, such as controllable summarization (Zhang et al., 2023). The flexibility of COLLIE allows these "functional constraints" to be incorporated for more usefulness, which we leave for future work.

**Grammar-based compositional tests.** Building benchmarks with data synthesized from grammars has been explored previously in the context of question answering (Weston et al., 2015), instruction following (Chevalier-Boisvert et al., 2019; Ruis et al., 2020)) and visual reasoning (Johnson et al., 2017). These benchmarks showcased the utility of grammars to systematically generate a comprehensive set of test cases or to specify some fixed constraints. In contrast, COLLIE aims to enable flexible, and dynamic constraint construction that can co-evolve with models. Furthermore, previous datasets were synthetic with limited linguistic diversity and practical applicability to real-world scenarios. In contrast, since our COLLIE framework extracts values and examples from natural language corpora to construct the constraints, it represents a more realistic challenge for modern LLMs.

**Systematic and scalable language benchmarks.** The emergence of increasingly powerful general-purpose language models has created a need for scalable benchmarks that can systematically and comprehensively evaluate them. A few recent examples include HELM (Liang et al., 2022), BIG-Bench (Srivastava et al., 2022), MMLU (Hendrycks et al., 2020), TaskBench500 (Li et al., 2022a), and Natural Instructions (Wang et al., 2022). However, building such benchmarks require considerable human effort, and may become obsolete when stronger models enter the arena. We provide a new perspective in this race between model capabilities and challenging benchmarks: leverage compositionality to construct automatic and scalable benchmarks with minimal human effort that can co-evolve with model capabilities to remain challenging and relevant.

## 3 COLLIE FRAMEWORK TO CONSTRUCT CONSTRAINED TEXT GENERATION

COLLIE allows researchers to easily 1) specify textual constraint structures via a grammar, then automatically 2) extract constraint values from text corpora, 3) render constraints into natural language instructions, and 4) evaluate generations with respect to constraints.

**Grammar.** Two observations about text constraints motivate a grammar characterization: 1) they involve different *levels* of text, e.g. character, word, sentence, or paragraph; and 2) many of them specify either the *count* or *position* at a certain text level (*existence* is equivalent to *count* $> 0$).

Let capitalized letters $(S, M, C, T)$ denote non-terminal variables, and other symbols $(\ell, \circ, \oplus, v)$ denote terminals. A full **constraint specification** within our grammar $S$ (Eq. 1) consists of two parts: a generation level ($\texttt{level}(\xi) = \ell$) specifying whether the generated text $\xi$ should be a word, a sentence, a paragraph, or a document, and a **multi-constraint** $M$ (Eq. 2), which is a logical composition of one or more **base-constraints** $C$. A **text** $T$ (Eq. 4) within these constraints can either be the full generated text $\xi$, or a part of it when qualified with a $\texttt{pos}(\cdot)$. For example, $\texttt{pos}(\texttt{pos}(\xi, \text{paragraph}, 3), \text{sentence}, -1)$ means "the last sentence of the 3rd paragraph of the generated text". For terminal variables, we define a **level** $\ell$ of a text (Eq. 5), a string or number **relation** $\circ$ or $\oplus$ (Eq. 6), and a string or number **value** $v_{\text{str}}$ or $v_{\text{num}}$ (Eq. 7). $\wedge$ represents the logical 'and' operator, and $\vee$ represents the logical 'or'. With these definitions, we construct the following grammar:

$$S \rightarrow (\texttt{level}(\xi) = \ell) \wedge M \qquad\qquad \text{(constraint specification)} \qquad (1)$$

$$M \rightarrow C \mid C \wedge M \mid C \vee M \qquad\qquad \text{(multi-constraint)} \qquad (2)$$

$$C \rightarrow \texttt{count}(T, \ell, v_{\text{str}} \mid \ell') \oplus v_{\text{num}} \mid \texttt{pos}(T, \ell, v_{\text{num}}) \circ v_{\text{str}} \qquad \text{(base-constraint)} \qquad (3)$$

$$T \rightarrow \xi \mid \texttt{pos}(T, \ell, v_{\text{num}}) \qquad\qquad \text{(text)} \qquad (4)$$

$$\ell \rightarrow \text{char} \mid \text{word} \mid \text{sentence} \mid \text{paragraph} \mid \text{passage} \qquad \text{(level)} \qquad (5)$$

$$\circ \rightarrow = \mid \neq \qquad \oplus \rightarrow = \mid \neq \mid > \mid < \mid \leq \mid \geq \qquad \text{(relation)} \qquad (6)$$

$$v_{\text{str}} \in \Sigma^* \qquad v_{\text{num}} \in \mathbb{Z} \qquad\qquad \text{(value)} \qquad (7)$$

At the core of our grammar, we consider two (symmetrical) types of base-constraints $C$ (Eq. 3):

**1. Count constraints.** $\texttt{count}(T, \ell, v_{\text{str}}) \oplus v_{\text{num}}$ constrains the occurrences of a particular level-$\ell$ string $v_{\text{str}}$. For example, $\texttt{count}(T, \text{word}, \text{'happy'}) \leq 3$ means "T should contain the word 'happy' no more than 3 times". In contrast, $\texttt{count}(T, \ell, \ell') \oplus v_{\text{num}}$ constrains the occurrences of level-$\ell$ strings in each level-$\ell'$ unit of text T. For example, $\texttt{count}(T, \text{char}, \text{sentence}) = 50$ means "each sentence of text T should have exactly 50 characters".

**2. Position constraints.** $\texttt{pos}(T, \ell, v_{\text{num}}) \circ v_{\text{str}}$ specifies that a particular part of the text T should equal (or not equal) the given string $v_{\text{str}}$. For example, $\texttt{pos}(T, \text{word}, 3) = \text{'happy'}$ means "the 3rd word should be 'happy' in text T". We also allow negative indices for reverse counting, e.g. $\texttt{pos}(T, \text{char}, -1) \neq \text{x}$ means "the last letter should not be 'x' in text T".

Note that the grammar above can easily be extended to accommodate more types of base-constraints (e.g. part of speech, sentiment) by implementing the corresponding semantic checks — we leave this to future work. Also for convenience, we use **constraint structure** to refer to a family of constraint specifications that only differ in their values (e.g. *generate a sentence with exactly x words*, $x \in \mathbb{N}$), and **constraint** to refer to a particular constraint specification with concrete values (e.g. *generate a sentence with exactly 5 words*).

**Examples and conceptual challenges.** Our grammar can express a wide range of constraints through logical compositions of base-constraints across different text levels. Table 1 illustrates some structures across generation levels, identified by names such as `para01` for paragraph generation, etc.

In addition to the generation levels, `count` and `pos` across different levels introduce a variety of challenges. For example, `word01` and `sent01` challenge token-based language models to count characters; `pass01` requires high-level semantic planning for models to generate a coherent passage under constraints; `sent04` and `para02` challenge models to generate text with presence or absence of particular words; `sent03`, `para03`, and `para04` require counting at multiple levels; and `word02`, `word03`, `sent02`, `para05`, and `pass01` combine counting and positional challenges at different levels, which can be considered most demanding conceptually. We empirically assess the difficulty of constraint structures in Section 5. Example COLLIE usage is presented in Figure 2.

```python
from collie.constraints import (
  Constraint, TargetLevel,
  Count, Relation
)
c = Constraint(
  target_level=TargetLevel('word'),
  transformation=Count(),
  relation=Relation('=='),
)
text = 'This is a good sentence.'
print(c.check(text, 5)) # True
```

Figure 2: Example COLLIE code for a simple number of words constraint.

Table 1: List of all constraint structures used in COLLIE-v1, with (simplified) example values.

| ID | Example instruction | Multi-constraint $M$ |
|---|---|---|
| word01 | Generate a word with at least 15 letters. | $\texttt{count}(\xi, \text{char}, \text{word}) \geq 15$ |
| word02 | Generate a word with 10 letters, where letter 1 is 's', letter 3 is 'r', letter 9 is 'e'. | $\texttt{count}(\xi, \text{char}, \text{word}) = 10 \wedge \texttt{pos}(\xi, \text{char}, 1) = \text{'s'} \wedge \texttt{pos}(\xi, \text{char}, 3) = \text{'r'} \wedge \texttt{pos}(\xi, \text{char}, 9) = \text{'e'}$ |
| word03 | Generate a word with at most 10 letters and ends with "r". | $\texttt{count}(\xi, \text{char}, \text{word}) \leq 10 \wedge \texttt{pos}(\xi, \text{char}, -1) = \text{'r'}$ |
| sent01 | Please generate a sentence with exactly 82 characters. Include whitespace into your character count. | $\texttt{count}(\xi, \text{char}, \text{sentence}) = 82$ |
| sent02 | Generate a sentence with 10 words, where word 3 is "soft" and word 7 is "beach" and word 10 is "math". | $\texttt{count}(\xi, \text{word}, \text{sentence}) = 10 \wedge \texttt{pos}(\xi, \text{word}, 3) = \text{"soft"} \wedge \texttt{pos}(\xi, \text{word}, 7) = \text{"beach"} \wedge \texttt{pos}(\xi, \text{word}, 10) = \text{"math"}$ |
| sent03 | Generate a sentence with at least 20 words, and each word less than six characters. | $\texttt{count}(\xi, \text{word}, \text{sentence}) \geq 20 \wedge \texttt{count}(\xi, \text{char}, \text{word}) \leq 6$ |
| sent04 | Generate a sentence but be sure to include the words "soft", "beach" and "math". | $\texttt{count}(\xi, \text{word}, \text{'soft'}) > 0 \wedge \texttt{count}(\xi, \text{word}, \text{'beach'}) > 0 \wedge \texttt{count}(\xi, \text{word}, \text{'math'}) > 0$ |
| para01 | Generate a paragraph where each sentence begins with the word "soft". | $\texttt{pos}(\texttt{pos}(\xi, \text{sentence}, 1), \text{word}, 1) = \text{'soft'} \wedge \texttt{pos}(\texttt{pos}(\xi, \text{sentence}, 2), \text{word}, 1) = \text{'soft'} \wedge ...$ |
| para02 | Generate a paragraph with at least 4 sentences, but do not use the words "the", "and" or "of". | $\texttt{count}(\xi, \text{sentence}, \text{paragraph}) \geq 4 \wedge \texttt{count}(\xi, \text{word}, \text{'the'}) = 0 \wedge \texttt{count}(\xi, \text{word}, \text{'and'}) = 0 \wedge \texttt{count}(\xi, \text{word}, \text{'of'}) = 0$ |
| para03 | Generate a paragraph with exactly 4 sentences, each with between 10 and 15 words. | $\texttt{count}(\xi, \text{sentence}, \text{paragraph}) = 4 \wedge \texttt{count}(\xi, \text{word}, \text{sentence}) \geq 10 \wedge \texttt{count}(\xi, \text{word}, \text{sentence}) \leq 15$ |
| para04 | Generate a paragraph with at least 3 sentences, each with at least 15 words. | $\texttt{count}(\xi, \text{sentence}, \text{paragraph}) \geq 3 \wedge \texttt{count}(\xi, \text{word}, \text{sentence}) \geq 15$ |
| para05 | Generate a paragraph with 2 sentences that end in "math" and "rock" respectively. | $\texttt{count}(\xi, \text{sentence}, \text{paragraph}) = 2 \wedge \texttt{pos}(\texttt{pos}(\xi, \text{sentence}, 1), \text{word}, -1) = \text{"math"} \wedge \texttt{pos}(\texttt{pos}(\xi, \text{sentence}, 2), \text{word}, -1) = \text{"rock"}$ |
| pass01 | Generate a passage with 2 paragraphs, each ending in "I sit." and "I cry." respectively. | $\texttt{count}(\xi, \text{paragraph}, \text{passage}) = 2 \wedge \texttt{pos}(\texttt{pos}(\xi, \text{paragraph}, 1), \text{sentence}, -1) = \text{"I sit."} \wedge \texttt{pos}(\texttt{pos}(\xi, \text{paragraph}, 2), \text{sentence}, -1) = \text{"I cry."}$ |

In conjunction with the grammar, we develop a set of *compiling tools* to help construct datasets with minimal human efforts. Concretely, the pipeline of dataset construction involves 4 stages (Figure 1):

**1. Specify constraint structures.** Researchers can specify constraint structures (e.g. Table 1), and optionally with a value range (e.g. "generate a sentence with $x$ words", and $5 \leq x \leq 10$). This is the only stage that involves manual effort.

**2. Extract constraint values from corpora.** We design an automatic extraction algorithm that runs through a given text corpus to find strings that fit a constraint structure with some value ranges. For example, given the constraint structure $\texttt{count}(\xi, \text{word}, \emptyset) = x$ with value range $5 \leq x \leq 10$, the extraction algorithm returns sentences in the corpus that have 5-10 words, with associated word counts. This ensures each constraint has at least one natural solution. More details are in Section 4.1.

**3. Render natural language instructions.** Each constraint can be rendered into a natural language instruction (Table 1) via ruled-based translation, thanks to the compositionality grammar of COLLIE. For example, a constraint $\texttt{count}(\xi, \text{char}, 'v') = 2 \wedge \texttt{count}(\xi, \text{char}, 'i') = 3$ can be synthetically rendered into the instruction "Please generate a word with exactly 2 character 'v' and exactly 3 character 'i'.". It is also possible to improve the instruction fluency or naturalness by adding additional rules to the synthetic translation, or use LLMs to polish instructions. More details are in Section A.1.

**4. Evaluate generations.** Given text $\xi$ generated by a model, we use a parser to evaluate it against a constraint specification $S$ and derive a True/False value, indicating if $\xi$ satisfies $S$. We use an average *success rate* as the main metric to evaluate constraint conformance. We can also compare the fluency of $\xi$ against the corpus-extracted "groundtruth" text, and render more fine-grained natural language feedback indicating which base-constraints are met and which not (see Section A.2).

## 4 COLLIE-v1 DATASET

We construct COLLIE-v1 using constraints structures from Table 1, which contains 2,080 constraint instances from 13 constraint types, with 1,435 unique constraint prompts. The broader NLP community can contribute to future dataset releases by adding additional constraints, metrics, data sources. The curated constraint set can co-evolve with models to become more challenging and comprehensive as model capabilities improve.

### 4.1 CONSTRAINT SPECIFICATION AND EXTRACTION

**Constraint specification.** We begin by defining 13 constraint structures. We chose these 13 structures to span various generation levels (word, sentence, paragraph and passage generation) and challenges (counting, position). In total, we have 3 word-level, 4 sentence-level, 5 paragraph-level, and 1 passage-level constraint structures. Of these 13 constraint structures, 5 are single-level and the remaining 8 are multi-level constraints. See Table 1 for the exact constraint structures we use.

**Constraint extraction.** While constructing constraint structures is straightforward using our grammar, choosing constraint targets is challenging for two reasons: (1) Not all targets will admit a conforming natural language string. For instance, the constraint, "Generate a two word sentence beginning with the word *The*." has no grammatically acceptable answer. (2) Even if a constraint admits a *possible* answer, it may not admit a *plausible* answer. For instance, "Generate a sentence with 1928 words" is possible, but any such sentence is very unlikely to appear in regular discourse.

To address both challenges, we sample constraint target values from natural language corpora, which we denote as the *data source*. Given a constraint structure $\mathcal{C}$ and documents $\mathcal{D} = \{d_1, ..., d_n\}$, we chunk each document into a series of strings $d_i = \{s_1, ..., s_m\}$, where each $s_i$ can be a sentence, paragraph, or passage as required by $\mathcal{C}$. Each string $s_i$ undergoes source-specific automated filtering and post-processing to remove artifacts, which we detail in Section B.4. Given $\mathcal{C}$ and $s_i$, we extract target values such that $\mathcal{C}$ is satisfied. In most cases, the satisfying target values can be directly extracted using our provided utilities. For example, for constraints with structure "sentence with $x$ words", we can directly apply word tokenization and counting to the example string $s_i$. In cases in which direct extraction is not possible, (e.g. "do not include word $w$"), we specify a range of possible targets (e.g. {*the, and, of* }) to sweep over. All in all, our approach ensures that (1) there exists a natural language string that can satisfy each constraint and target pair, and (2) the targets follow a

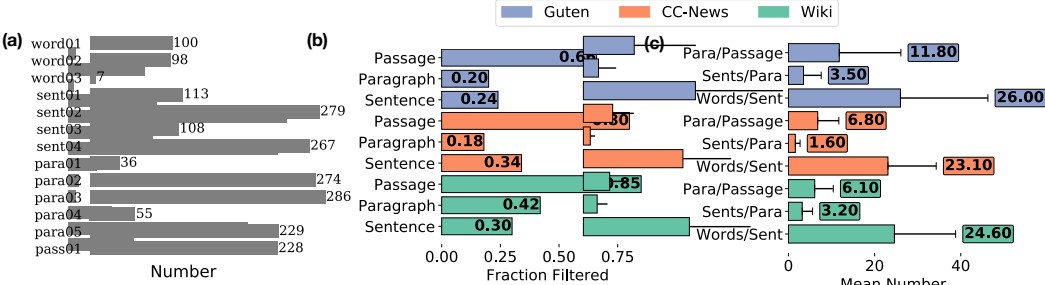

Figure 3: **Data statistics. (a)** Number of constraints from each constraint structure. **(b)** Fraction of strings removed by automated filtering. **(c)** Length statistics for different levels for each data source.

plausible distribution induced by natural language corpora. Our extraction system is extensible, and can operate on new constraints and data sources with minimal modifications.

**Extensibility** Adding additional data sources to the extraction pipeline is similarly easy, requiring a text delimiter, and optional string filtering and post-processing functions. As a case-study on the extensibility of COLLIE, we demonstrate how to extend constraints to include POS-tags such as "Generate a sentence with verbs". Details are in section B.5.

## 4.2 DATA SOURCES

To adequately cover diverse styles and content, we extract constraint targets from three distinct data sources: *Wikipedia* (Wiki) (Foundation, 2022), *Common Crawl News* (CC-News) (Hamborg et al., 2017), and the *Project Gutenberg Corpus* (Guten) (Brooke et al., 2015). We provide an overview of these data sources below and leave source-specific filtering and post-processing details to Section B.4.

**Wiki.** Wikipedia (Wiki) (Foundation, 2022) consists of over 6 million English Wikipedia articles. We included this data source for the diverse subject matter present in the corpus.

**CC-News.** The Common Crawl News corpus (CC-News) (Hamborg et al., 2017) consists of 708,241 English language news articles published between Jan 2017 and December 2019. We include CC-News to include interview dialogues, as well as popular culture and current events.

**Guten.** The *Project Gutenberg* corpus (Guten) (Brooke et al., 2015) consists of over 50,000 documents that include fiction, histories, biographies, and other works that are in the public domain in the United States. We include this corpus for its variety in genres (e.g. non-fiction, fiction, plays, etc.) and style from different time periods.

## 4.3 DATA VALIDATION AND STATISTICS

We extract constraints from 300 randomly sampled documents from each source. After extracting the target values, we sample up to 100 targets for each constraint structure on each data source. We remove any string targets by that begins or ends with any character that is not a letter or number. We randomly sample 5 out of these 100 targets and their supporting examples to qualitatively verify their validity. Since the extraction process is relatively fast, we modify filters and post-processors if there are systemic issues and re-run the extraction phase. We provide statistics of the final number of constraints from each constraint structure in Figure 3(a). Some constraints (e.g. number of sentences per paragraph) are tightly clustered around the mean, and thus do not induce many valid constraint targets. The fraction of strings filtered for each data source and level is presented in Figure 3(b). The automated filtering removes a large fraction of the strings in most cases, as high recall is important to ensure quality. The high fraction of omitted passages is due to the removal of passages < 2 paragraphs in length. Mean lengths for each level and data source is presented in Figure 3(c).

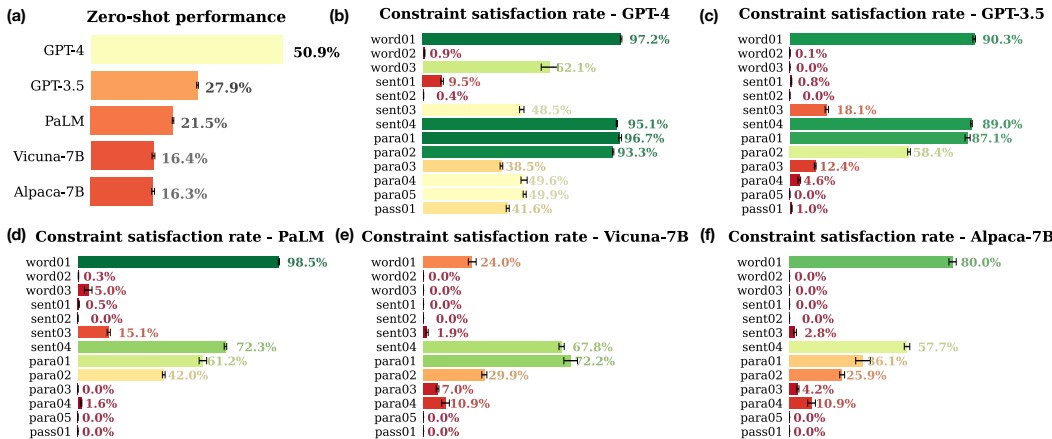

Figure 4: **Model comparison. (a)** Overall model performance summarized by weighted average across all constraint groups. **(b)** -**(f)** Constraint satisfaction rates of generated texts by GPT-4, GPT-3.5, PaLM, Vicuna-7B, and Alpaca-7B across various constraint groups. Error bars represent standard error. Constraint group names are in Table 1. Sample sizes are reported in Figure 11.

## 5 RESULTS

Our main experiments in this paper focus on a zero-shot prompting setup with the following language models (LMs): 1) larger and closed-source LMs such as OpenAI GPT (Brown et al., 2020; OpenAI, 2023b) (gpt-3.5-turbo, gpt-4) and Google PaLM-2 (Anil et al., 2023) (text-bison-001); 2) smaller and open-source LMs such as Alpaca-7B (Taori et al., 2023), Vicuna-7B (Chiang et al., 2023). We performed additional one-shot prompting and find GPT performances similar to zero-shot performance, see Section C.1. By default, we use a sampling temperature of 0.7, and sample multiple trials (20 for GPT/PaLM, 5 for Alpaca/Vicuna). All experiments were run in July, 2023.

**Zero-shot performance comparison.** As evidenced in Figures 4(a), GPT-4 consistently surpassed other models in zero-shot constrained text generation performances, achieving more than twice the constraint satisfaction rate than other non-GPT models. The overarching performance trend observed shows GPT-4 leading the pack, followed by GPT-3.5 and PaLM with a large gap, and then followed closely by the smaller models, Vicuna-7B and Alpaca-7B.

**Constraints all models can follow.** Certain tasks, specifically word01 (generating a word with at least $a$ letters), sent04 (generating a sentence containing words X, Y, Z), and para01 (generating a paragraph with each sentence starting with the word X), posed minimal challenge to the majority of contemporary language models. These tasks demonstrate the proficiency of current models at simple constraints ensuring existence, as depicted in Figure 9(f).

**Constraints partially solved by GPT-4 only.** However, a notable distinction arose when tasks incorporated more counting/position constraints and requested longer generations. Tasks such as word03, para04, para05, and pass01 were only partially addressed by GPT-4, with constraint satisfaction rates ranging between 40% and 70%. Despite GPT-4's partial success in these tasks, other models failed to deliver any satisfactory performance.

**Constraints remaining very challenging.** Furthermore, some tasks proved challenging across all models. Tasks word02, sent01, sent02, and para03 present challenges in terms of arbitrary position constraints and mixed counting levels (see Section 5.1 for detailed analysis), indicating areas that necessitate further advancements in language model technology. Moreover, the average pass@20 rate of GPT-4 was above 63% across all constraints, significantly higher than the 32% achieved by GPT-3.5, as depicted in Figure 5. Although GPT-4 demonstrated a significant performance advantage, its constraint satisfaction rate of 63% is far from perfect. This suggests considerable scope for improvement in controllable text generation with language models. These findings underscore the opportunities and challenges in the continued evolution of language models.

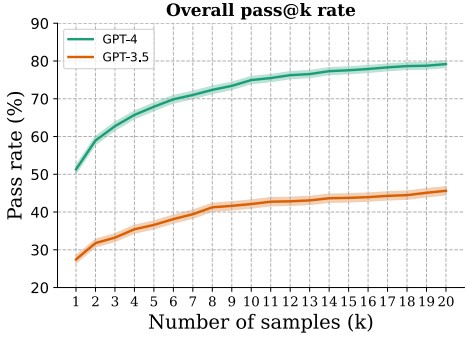

Figure 5: **Pass@k performance.** We sample the model-generated text 20 times for all instruction prompts in the dataset. The curves represent the average pass rate across all instruction prompts up to $k$ samples. The shaded areas indicate the standard errors.

Figure 6: **Position effect.** Satisfaction rates of LMs on tasks involving $\mathrm{pos}(\xi, \mathrm{level}, i)$. The tasks word02 and sent02 impose constraints on characters and words at arbitrary positions. The task para01 constrains the first word. The tasks word03, para05, and pass01 constrain the last characters, words, and sentences.

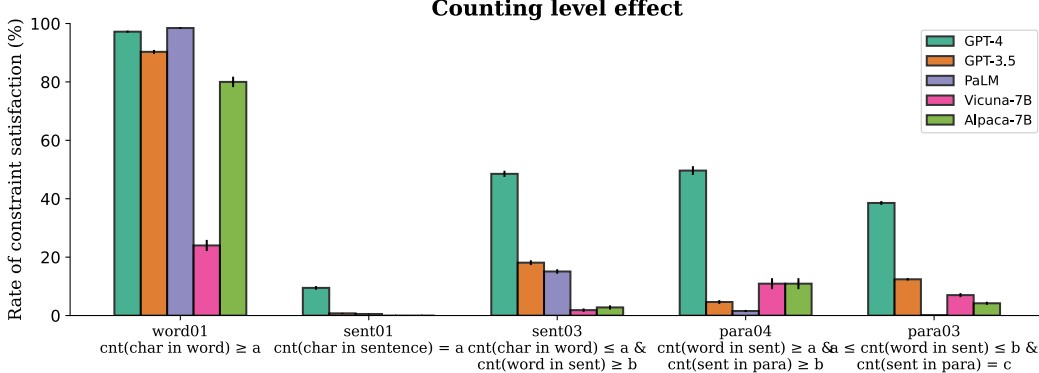

Figure 7: **Counting level effect.** Satisfaction rates for LMs on tasks involving $\mathrm{count}(\xi, \mathrm{level}, \varphi)$. Task word01 sets a minimum word length of $a$. Task sent01 requires exactly $a$ characters in a sentence. Task sent03 asks a sentence to contain at least $b$ words, with each word no longer than $a$ letters. Task para04 asks a paragraph to consist of at least $b$ sentences, each containing a minimum of $a$ words. Task para03 further imposes an upper limit on the number of words per sentence.

## 5.1 ANALYSIS

**Performance consistency across data sources.** We observe a high degree of consistency in the performance of models on a given constraint structure, regardless of the data source. This uniformity is evident across all models, as highlighted in Figure 9 (g). This indicates that the ability of a language model to adhere to the logic of constraints takes precedence over the specific target values or the distribution of the data.

**Position effect.** As depicted in Figure 6, the $\mathrm{pos}(\xi, \mathrm{level}, i)$ function, constraining the $i$-th sub-string (letter, word, or sentence), exhibits varying levels of difficulty depending on the value of $i$. Models generally perform well when the positional constraint is applied to the first sub-string ($i = 1$, task para01). However, only GPT-4 displays partial success with the last positional constraints ($i = -1$, tasks word03, para05, pass01). Notably, all models encounter difficulties when generating text that satisfies positional constraints at arbitrary positions $i$. Additionally, we find that the position effect exhibits a lower sensitivity to constraint levels.

**Counting level effect.** Counting characters within a word is easier than within a sentence for models, as illustrated in Figure 7. Furthermore, tasks demanding exact equality (task sent01) prove more

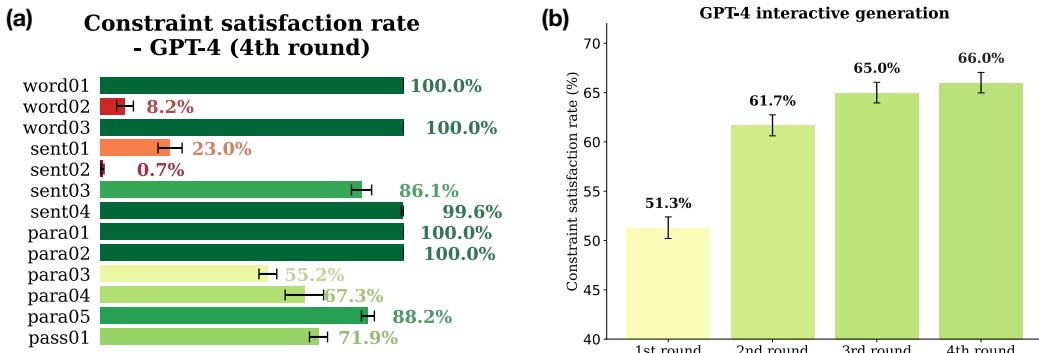

Figure 8: **GPT-4 interactive generation performance.** **(a)** Constraint satisfaction rate of GPT-4 generated texts in the 4th round across various constraint groups. **(b)** GPT-4 overall performance in different feedback rounds. The 1st round is zero-shot, and the 2nd - 4th rounds are with feedback.

challenging than those requiring a range (task `para03`), and are considerably more difficult than tasks specifying just an upper or lower bound (tasks `word01`, `sent03`, `para04`).

**Increased difficulty with logical composition.** The incorporation of logical compositions into constraints considerably increases their difficulty. Task `sent03` serves as an example of this, adding an extra constraint at the sentence level compared to task `word01`. Despite the assumption that the added constraint should be manageable for all models, performance on task `sent03` uniformly trails behind that on task `word01`, as shown in Figure 7. This highlights the intricacy and challenge introduced by logical compositions within constraints.

**Performance enhancement through feedback and interaction.** We utilize COLLIE to generate automated natural language feedback (e.g., "Your task is to generate a word with exactly 2 character 'v' and exactly 3 character 'i'. However, you generate a word with 3 character 'v' and 4 character 'i'."), and engage LLMs in a generation-feedback dialogue. In Figure 8, we observe a significant 20% improvement in GPT-4 performance after the second round of feedback. However, the model's performance plateaus at 66% even after three additional rounds of feedback, comparable to pass@5 using i.i.d. sampling. The extent of performance improvement varies across tasks, with `word03`'s constraint satisfaction rate increasing from 62.1% to 10%. Conversely, `word02`, `sent01`, and `sent02` tasks remain challenging for the model. These findings suggest that there is still room for improvement, highlighting the difficulty of our dataset, and emphasizing the need for further research on better ways to incorporate natural language feedback.

## 6 CONCLUSION

In this work, we present COLLIE, a grammar-based framework for specifying textual constraints. COLLIE simplifies the process of creating constrained-generation datasets by enabling researchers to focus on specifying high level constraint structures, while COLLIE automatically extracts constraint values, renders natural language instructions, and assesses model performance. To demonstrate the utility of the COLLIE framework, we construct COLLIE-v1 with 1,132 constraints from 13 different types, extracted from 3 different data sources. We evaluate five state-of-the-art LLMs of various sizes on COLLIE-v1, and find that it provides fine-grained insights into model capabilities and shortcomings. We hope that model developers can use COLLIE-v1 to develop more capable models, while future releases of COLLIE can continue to adapt to the capabilities and needs of future models and users.

## LIMITATIONS AND SOCIETAL IMPACTS

Although care was taken to design the filtering and processing functions, such automated approaches are never perfect and remaining artifacts in corpora might lead to unnatural reference texts or constraints. Further filtering (e.g., by grammar checkers, parsers, or humans) could improve the dataset quality. Our representative constraint structures were selected to encompass diverse constrained generation challenges, but as with all generation benchmarks, they cannot capture all dimensions

and nuances of model capabilities. Benchmarks are highly influential in shaping model development, the capabilities and limitations of which may disproportionately impact different communities. Our benchmark is no exception. However, by providing an extensible, easy-to-use framework for constraint development, we hope COLLIE will enable diverse stakeholders to engage with dataset building, helping ensure that future model capabilities serve diverse interests and needs.

## ACKNOWLEDGEMENTS

We thank Xiao Liu for Vicuna/Alpaca APIs that supported our preliminary experiments, and Princeton NLP Group for helpful discussion and feedback in general. We acknowledge support from a Princeton SEAS Innovation grant and the National Science Foundation under Grant No. 2239363. Any opinions, findings, and conclusions or recommendations expressed in this material are those of the author(s) and do not necessarily reflect the views of the National Science Foundation.

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
