## A    NATURAL LANGUAGE RENDERING OF CONSTRAINTS

### A.1    INSTRUCTION RENDERING

COLLIE provides a rule-based constraint renderer that converts constraints into natural language instructions (see examples in Table 1).

Leveraging the compositionality of the context-free grammar, the renderer first parses the constraint as a tree. In the case of multi-constraints, it generates prompts for each base-constraint individually and then concatenates the generated prompts together at the end. For a base-constraint, it follows a pre-order traversal of the subtree to modify the initial template "Please generate a {generation_level} with @... {tagert_level}," where "@..." serves as a placeholder.

Although the rule-based instruction prompts are natural enough for all examples in COLLIE-v1, there might be some edge cases where the rule-based instructions are not fluent enough for newly specified constraints. To address this, we offer an option to utilize language models to enhance the rule-based instructions. We employ the following prompt for the language model to refine the instructions: "Please rewrite the following paragraph to improve fluency without altering the original meaning. You should provide the revised paragraph directly. Original paragraph: {prompt}."

The renderer is independent of the constraint construction and can be easily extended with new rules for parsing and mapping to instruction templates.

### A.2    FEEDBACK RENDERING

We further extend our framework to provide natural language feedback when the extracted value of the generated text differs from the target value. Similar to instruction rendering, we first employ a rule-based renderer to compose the feedback by modifying a template. We also provide an option to use language models to polish the generated feedback.

For instance, consider a constraint $\texttt{count}(\xi, \text{char}, 'v') = 2 \wedge \texttt{count}(\xi, \text{char}, 'i') = 3$, while the generated word includes three 'v' and four 'i'. Our framework can generate the following instructions and feedback:

- INSTRUCTION:
  Please generate a word with exactly 2 character 'v' and exactly 3 character 'i'.

- GPT-POLISHED INSTRUCTION:
  Please generate a word that contains exactly 2 instances of the letter 'v' and exactly 3 instances of the letter 'i'.

- FEEDBACK:
  Your task is to generate a word with exactly 2 character 'v' and exactly 3 character 'i'. However, you generate a word with 3 character 'v' and 4 character 'i'.

- GPT-POLISHED FEEDBACK:
  Your task was to generate a word with precisely 2 'v' characters and precisely 3 'i' characters. However, you generated a word with precisely 3 'v' characters and precisely 4 'i' characters.

By incorporating this feedback mechanism, our framework can provide explicit guidance for the language models to improve the generation quality and adhere to the specified constraints.

## B    DATASET

### B.1    EXTRACTION OVERVIEW

The extraction phase is split into six steps.:

1. **Document loading.** The document $d$, usually consisting of multiple paragraphs is loaded.
2. **Text chunking.** Each document is divided into paragraphs using a source-specific delimiter (e.g. \n). To obtain sentences, we use the nltk (Bird, 2006) sentence tokenizer on each paragraph.

To obtain passages, we string together multiple consecutive paragraphs that survive filtering. To obtain word-level constraint targets, we iterate each $s_i$ from an English language word list.

3. **Text filtering.** The paragraph or sentence passes a source-specific filtering function that attempts to remove all strings that are not natural language, for instance copyright statements.

4. **Text post-processing.** The paragraphs or sentences that survive filtering are post-processed to remove source-specific artifacts, such as Markdown formatting.

5. **Passage construction (passage-level only).** Paragraphs and sentences pass through to the next step. Passages are constructed by appending as many consecutive paragraphs that survive filtering as possible. For instance, if a document contains paragraphs $p_1, ..., p_9$, and $p_4$ is the only paragraph that is removed due to filtering, then we return two passages: $(p_1, p_2, p_3)$ and $(p_5, ..., p_9)$. Each paragraph is joined by two newline characters within each passage.

6. **Constraint extraction.** The sentence, paragraph, or passage-level string is passed to the constraint extractor that pulls out constraint targets from the string. This can either be done directly, such as directly extracting the total word count, or sweeping over a set of possible target values.

For each data source, we randomly sample 300 documents. For each document, we randomly sample up to 100 text sequences of the specified level (sentence, paragraph, or passage) for constraint extraction to prevent over-representation from very long documents. We then randomly sample up to 100 constraint targets for each constraint structure and data source. We now discuss source-specific details below:

## B.2 Text filters

In this section, we describe the text filter heuristics in detail. Note that which filters to use are source-specific.

- **URL.** This filter removes any string that contains a pattern that appears to be a URL. The pattern we find is expressed using the following regex:
  ```
  r"(http(s)?://)?(www\.)?[a-zA-Z0-9\-]+\.[a-zA-Z]{2,6}
  (\.[a-zA-Z]{2,6})?(/[a-zA-Z0-9\-]*)*(\?[a-zA-Z0-9\-=&]*)?"
  ```

- **All caps.** This filter removes any text that only contains capitalized letters, which may be indicative of a section heading.

- **No sentences.** This is a filter that tries to detect strings without any valid sentences in the text. We first sentence tokenize the string. If no "sentence" contains a period and has length greater than 2, then we remove the string. Otherwise, we keep it. Future improvements could use a parser or trained classifier.

- **Copyright.** This filter removes copyright statements typically found at the end of articles. It removes any string that contains the copyright symbol "©" or where the uncased first word is "copyright".

- **Caption.** This filter attempts to remove captions, such as those under diagrams or images. These strings typically follow the format: "Photo: a green car.". We heuristically detect such strings by rejecting any string where the number of characters to the left of the first ":" is less than six characters.

## B.3 Text post-processing

In this section, we describe the post-processing functions used on the strings. Note that which post-processing functions to use depends on the data-source.

- **Markdown removal.** We remove markdown artifacts using the following substitution rule: `(r'(\*\*|__|\*|_|backslash~\~)(.*?)\1', r'\2')`

- **Consecutive whitespace.** Consecutive whitespace is removed with the following substitution rule: `(r'\s{2,}', ' ')`

- **Single newline to space.** Single newlines are replaced with a single space using the following substitution rule: `(r"(?<!\n)\n(?!\n)", " ")`

- **Bracket removal.** We remove brackets from the text using the following substitution rule: `(r'\[[^\]]*\]',  "")`. This is useful for removing references inside the text.

## B.4   DATA SOURCES

Detailed statistics on the number of constraints extracted for each constraint structure for the grouped, and individual data sources are found in Figures 11 and 10 respectively. None of the datasets used contain PII, as far as authors are aware.

**Wiki**   We use the `20220301.en` train split of the dataset from Huggingface Lhoest et al. (2021). We split each document into paragraphs using two newlines as the delimiter (`\n\n`). We use three filters for Wiki: URL, caption, and no sentences. For our passage level constraint, we also omit any text that contains the vertical line character "|", as these were identified to often be tables. We use a Wiki specific post-processing function that removes any text before the first newline character, for any text that contains a newline character. We found that these are almost always section headings. Wiki is licensed under a CC BY-SA 3.0 license and GNU Free Documentation License[*].

**CC-News**   We load from the train split of the `cc_news` dataset on Huggingface Datasets for convenience. We split each document into paragraphs using a single newline as the delimiter (`\n`. We use four filters for CC-News: copyright, URL, cpation, and no sentences. We do not use any post-processing function. The TOS for this data can be found at `https://commoncrawl.org/terms-of-use/full/`.

**Guten**   We use the processed dataset from *Gutenberg, dammit*: `https://github.com/aparrish/gutenberg-dammit`. We split each document using two newlines as the delimiter (`\n\n`). We two filters for Guten: all caps, and no sentences. We post-process the text using four processors, applied in the following order: markdown removal, bracket removal, single newline to space, consecutive whitespace to single whitespace. All documents in Guten are in the public domain in the U.S.

**Words**   For word-level constraints, we iterate over the the words present in the following newline-separated word list: `http://www.gwicks.net/textlists/english3.zip`. We conduct no filtering or post-processing on the words from the list.

Our entire code, including those used for data extraction will be released under an MIT license.

---

[*]See `https://dumps.wikimedia.org/legal.html`

## B.5 EXTENDING COLLIE FOR POS-TAGS

To showcase the flexibility of Collie to incorporate new concepts for constraints, we updated our repo to allow constraints on POS tags. For example, "Generate a sentence with verbs" can be written as:

```
c = Constraint(
    target_level=TargetLevel('word'),
    transformation=Count('VERB'),  # specify target to be VERB
    relation=Relation('=='),
    attribute='pos',  # invoke the POS tagging model
)
# s has 2 verbs
s= 'Apple is looking at buying U.K. startup for \$1 billion.'
# True, because s has 2 verbs
print(c.check(s, 2))
# False, because s does not have only 1 verb
print(c.check(s, 1))
```

"Generate a sentence with the third word being verb" can be written as:

```
c = Constraint(
    target_level=TargetLevel('word'),
    transformation=Position(2), # specify the position
    relation=Relation('=='),
    attribute='pos',  # invoke the POS tagging model
)
# 3rd word is "looking", verb
s= 'Apple is looking at buying U.K. startup for $1 billion.'
# True, because the third word of s is verb
print(c.check(s, 'VERB'))
# 3rd word is "at", not verb
s= 'Apple looked at buying U.K. startup for $1 billion.'
# False, because the third word of s is not verb
print(c.check(s, 'VERB'))
```

Once the attribute argument is specified as `pos`, it invokes the POS tagging model in the constraint class to first tag the text at the target level. In a similar pattern, other attributes such as sentiment of the sentences in a passage can also be incorporated in a similar fashion by specifying the attribute argument to sentiment and invoke a sentiment classifier on the target level (e.g., sentence).

## C ADDITIONAL EXPERIMENTAL RESULTS

We note that out of 1132 constraints, 2 constraint prompts are blocked by PaLM-2 API for the guardrailing reason:

1. In `ccnews_c07`: Please generate a sentence containing the word 'charged', 'been', 'Father'.

2. In `ccnews_c14`: Please generate a passage with all paragraphs having the last sentence to be 'Gramercy's portfolio looks attractive relative to peers', 'I am going to add Gramercy Property Trust to my income portfolio this week', 'An investment in GPT yields 6.6 percent', 'The REIT's shares have slumped a whopping $\sim 15$ percent in 2018, but are no longer oversold' respectively. Only generate the passage, without extra things like "Paragraph 1" or "Answer:".

## C.1 ONE-SHOT EXPERIMENTS

To understand if LLM performances are bottlenecked by the zero-shot instruction format and if example input-output pairs could boost performances, we did a preliminary one-shot prompting experiment using an internal version of the dataset before the current COLLIE-v1, where for each constraint structure, we use a fixed constraint and its corpus example as an example input-output pair attached before the constraint prompt. As shown below in Table 2, results are very similar for both GPT-3.5 and GPT-4, which suggests the task difficulty is mainly about generation under the constraint instead of understanding the constraint (by similar examples).

Table 2: 0 vs. 1-shot results across all constraints (%).

|        | GPT-3.5 | GPT-4 |
|--------|---------|-------|
| 0-shot | 23.1    | 40.7  |
| 1-shot | 23.6    | 39.4  |

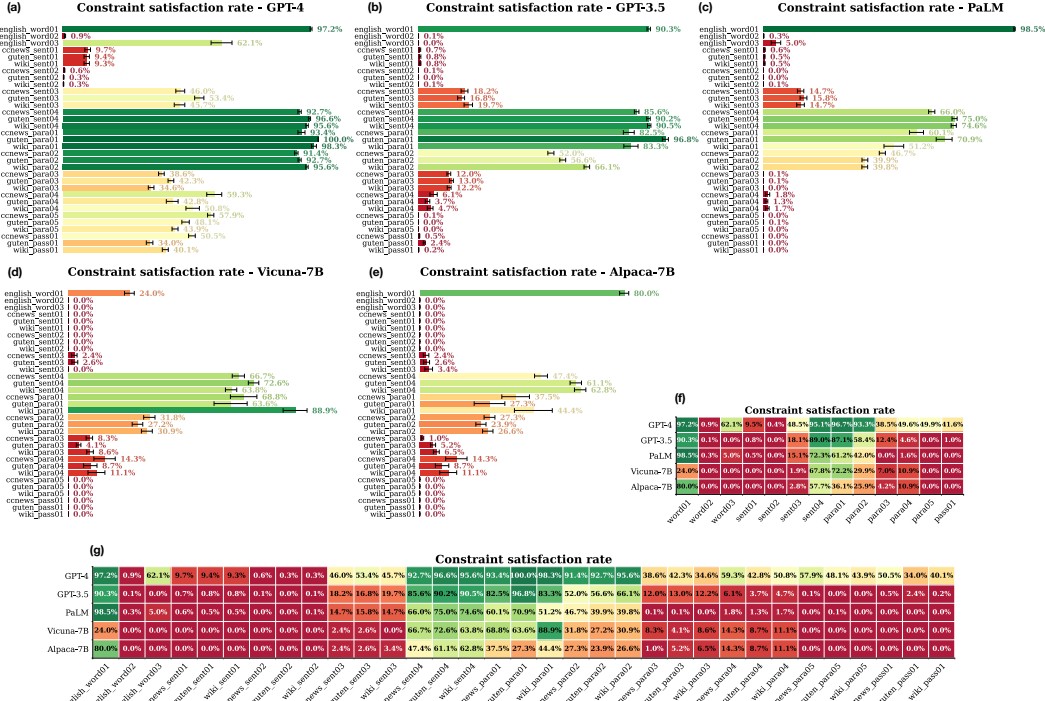

Figure 9: **Model performance on different constraints and datasets. (a)-(e)** Constraint satisfaction rates of texts generated by GPT-4, GPT-3.5, PaLM, Vicuna-7B, and Alpaca-7B across various constraints and datasets. Error bars indicate standard error. The constraint group names can be found in Table 1. Sample sizes are reported in supplementary Figure 10. **(f)** Summary heatmap of model performance on different constraint groups. **(g)** Summary heatmap of model performance on different constraints and datasets.

## C.2 CONSTRAINT SATISFACTION RATES

Figure 9 provides a comparison of constraint satisfaction rates for various models across all tasks. The performance of the models remains consistently high for a specific constraint structure, regardless of the data source. The satisfaction rates are summarized in heatmap Figures 9(f)-(g).

Figures 10 and 11 provide detailed information about the dataset size and sample size for each model in the study. Specifically, we conducted 20 trials for each instruction prompt in the case of GPT-4 and GPT-3.5. For PaLM, a total of 30 trials were conducted for each instruction prompt. However, due to a certain failure rate, the number of generated texts may not be a multiple of the number of instruction prompts. Vicuna-7B and Alpaca-7B were each run for 10 trials, and they also experienced some low failure rates during the experiments.

## C.3 ADDITIONAL EVALUATIONS

In addition to evaluating binary constraint satisfaction in general, it is also possible to evaluate particular aspects of text generation with respect to the constraint and extracted corpus text.

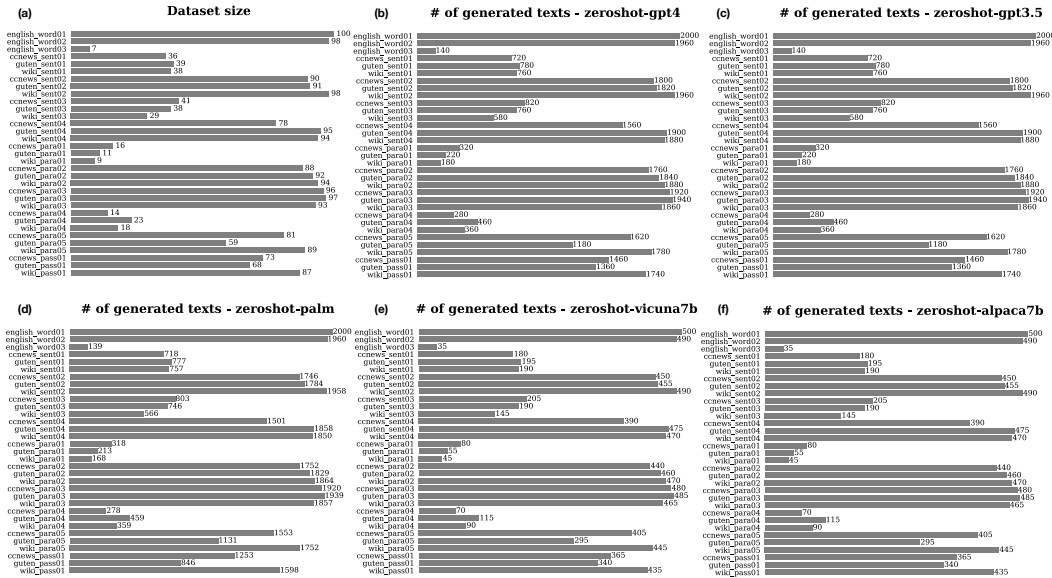

Figure 10: **Dataset and sample sizes.** **(a)** Dataset sizes for each constraint and data source. **(b)-(f)** Total sample sizes of generated texts from different models for each constraint and data source.

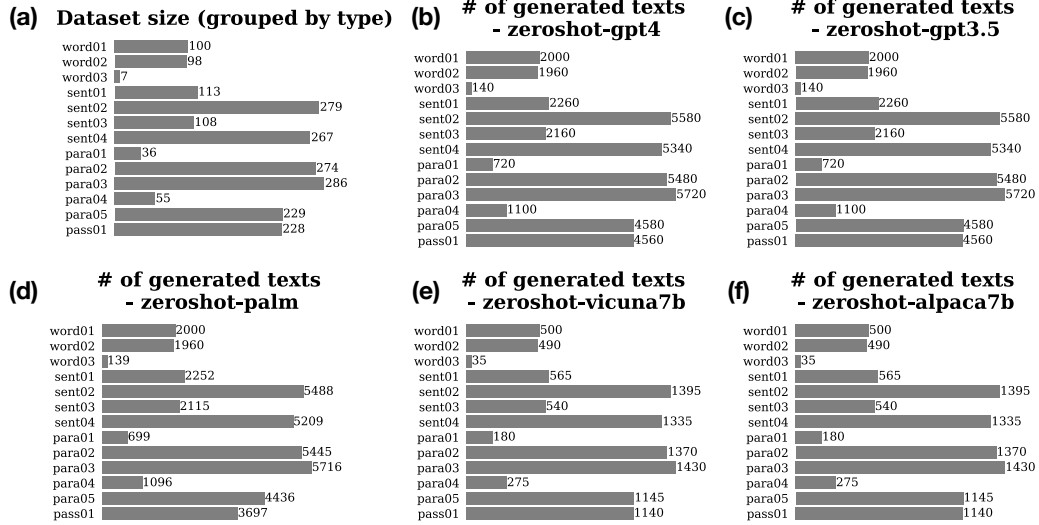

Figure 11: **Grouped dataset and sample sizes.** **(a)** Dataset sizes for each constraint group. **(b)-(f)** Total sample sizes of generated texts from different models for each constraint group.

**Word validity evaluation.** To determine the validity of word-level generations as English words, we cross-reference the generated words with the word list available at `http://www.gwicks.net/textlists/english3.zip`. Since this word list is not complete, we supplement it by including eight additional uncommon but valid English words: 'supercalifragilisticexpialidocious', 'pneumonoultramicroscopicsilicovolcanoconiosis', 'antidisestablishmentarianism', 'pseudopseudohypoparathyroidism', 'extraterrestrializationism', 'acceleratrix', 'circumlocutrix', and 'procrastinatrix'.

Figure 12 illustrates the performance comparison of different models in generating long words (`word01`). Notably, GPT-4 demonstrates superior performance compared to other models in generating long words. However, when faced with more challenging constraints, such as the requirement for the $i$-th letter to be 'r', all models fail to generate a word that satisfies the constraint (see Figure 9). In

this case, GPT-3.5 manages to generate valid words, while GPT-4 resorts to fabricating words like "coordinasor" to better conform to the constraints.

Regarding task `word03`, only GPT-4 is capable of generating words that satisfy the constraint on the last character. However, it still frequently generates made-up words. None of the other models are able to generate valid words or strings that satisfy the given constraint.

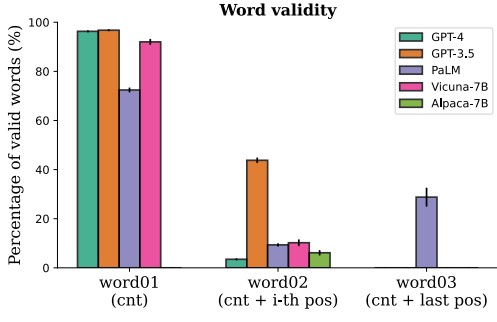

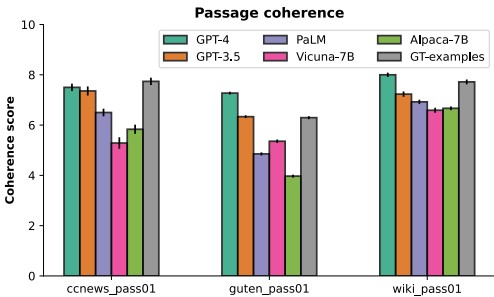

Figure 12: **Word validity.** Percentage of generated words that are "valid" words for a given English vocabulary list. Language models can sometimes generate plausible words, such as "co-ordinasor" and "adventudposis", but those are not common or valid words in modern English.

Figure 13: **Passage coherence.** Average passage coherence scores rated by GPT-4. Each generated passage was evaluated through three independent runs, while roughly one trial was taken for each model. GT-examples are ground truth. The error bars represent the standard error across the dataset.

**Passage coherence evaluation.** In order to assess the coherence and flow of content within the generated paragraphs, we utilize GPT-4 as a third-party judge to provide coherence scores. For this evaluation, we employ the following prompt: "Analyze the following passage, then conclude with the statement 'Thus, the coherency score is $s$,' where $s$ is an integer ranging from 1 to 10." We conduct three separate samplings of coherence scores for each generated text and calculate the average score. This methodology allows us to quantitatively measure the overall coherence of the generated paragraphs and gauge their coherence in a relatively consistent and reliable manner.

Figure 13 presents the coherence scores of generated passages for task `pass01`. Notably, both GPT-4 and GPT-3.5 consistently outperform the other models in terms of coherence. Furthermore, GPT-4 achieves a level of coherence that is comparable to the ground truth passages in the dataset.