# OpenReview forum: "COLLIE: Systematic Construction of Constrained Text Generation Tasks"
_ICLR.cc/2024/Conference — ICLR 2024 poster_

### Official Review · Reviewer_p51c · 2023-10-28

**Soundness:** 3 good
**Presentation:** 3 good
**Contribution:** 3 good
**Rating:** 6
**Confidence:** 4

**Summary:**

The authors propose COLLIE, a grammar-based framework for probing the performances of LLMs on constrained text generation tasks. COLLIE realizes the constraints on different levels (word, paragraph, passage) and imposes requirements on various aspects, e.g., character count, word count, and positions of a specific text. Experiments are conducted on instructions built from three sources and on various strong LLMs (GPTs, PaLMs, and etc.). The results suggest that GPT-4 generally outperforms other compared models but still performs poorly on a subset of the tasks.

Overall, while the concept of this task is simple, the challenges posed to existing LLMs are non-trivial, clearly pointing out the incapabilities of these models. I believe COLLIE has the potential to drive further developments of existing models on more nuanced tasks.

**Strengths:**

* The paper is well-written and easy-to-understand.
* COLLIE is a conceptually simple but scientifically non-trivial method, and is beneficial to further development of LLMs.
* The experiments and analyses are comprehensive, including evaluations on current state-of-the-art models.
* The authors also provide complementary code which helps verify their approach.

**Weaknesses:**

I have one question regarding to the instructions obtained from different sources. Since the grammar is translated into natural language instruction to prompt the model, how do the instructions from different source datasets differ from one another?

**Questions:**

See weakness section.

---

> ### Author Response · Authors · 2023-11-16
> **Thanks**
>
> The format of the instructions (i.e., constraint types) are identical across different sources, e.g., "Generate a passage with $n$ sentences."
>
> However, the distribution of constraint values (e.g., $n$ in the previous constraint) differ for different sources. For instance, Wiki has twice the average number of sentences compared to CC-News (See Fig. 3(c)).
>
> Let us know if you have further questions, thanks!

---

### Official Review · Reviewer_6Jty · 2023-10-29

**Soundness:** 3 good
**Presentation:** 3 good
**Contribution:** 3 good
**Rating:** 8
**Confidence:** 3

**Summary:**

This paper presents a framework, COLLIE, that allows researchers to build constrained text generation benchmark using different combinations of generation levels and modeling challenges (tasks), and a benchmark, COLLIE-v1, which is constructed using that framework and consists of 20,80 data instances comprising 13 structures.
The value of this paper lies in that it provides a method which allows future work to construct data of their interest in a scalable manner, and the analyses that the this paper conducts provide insights to researchers who are focusing on developing LLMs with better logical, reasoning, and compositional capacities.

**Strengths:**

1. This paper provides a method which allows future work to construct data of their interest in a scalable manner.
2. The analyses that the this paper conducts provide insights to researchers who are focusing on developing LLMs with better logical, reasoning, and compositional capacities.
3. This paper is generally well-written and easy to follow.

**Weaknesses:**

There is no great weaknesses that I can find - but there is a minor one:

Although I understand that the authors are focusing on more "basic" units, such as tokens, sentence, etc, so this paper can be more practical and useful for downstream applications, such as pretraining and evaluating LLMs, most of current work on constrained text generation seem to focus on text summarization, including controllable text summarization (e.g., MACSum Zhang et al, 2023). However this paper does not mention any of this, and does not discuss any potential that the proposed method can be applied to those constrained text generation (summarization) tasks. I personally suggest that the authors provide some discussion or explain on this issue.

Zhang, Yusen, Yang Liu, Ziyi Yang, Yuwei Fang, Yulong Chen, Dragomir Radev, Chenguang Zhu, Michael Zeng, and Rui Zhang. "Macsum: Controllable summarization with mixed attributes." Transactions of the Association for Computational Linguistics 11 (2023): 787-803.

**Questions:**

Please see weakness

---

> ### Author Response · Authors · 2023-11-16
> **Thanks**
>
> Thanks for your great suggestion! We have incorporated the related work and discussion in the draft!

---

### Official Review · Reviewer_PuAQ · 2023-11-01

**Soundness:** 3 good
**Presentation:** 3 good
**Contribution:** 2 fair
**Rating:** 6
**Confidence:** 3

**Summary:**

The authors propose a framework to automatically extract samples from large unlabeled text corpora for constrained text generation. Specifically, they manually craft rules/constraints and find satisfying texts as references. For example, a constraint can be "the third last word being mankind". They then evaluate off-the-shelf LLMs on this extracted dataset.

**Strengths:**

1. The authors evaluated different off-the-shelf LLMs and showed that they don't fully solve this task.

**Weaknesses:**

1. The idea is not very novel, and similar ideas have already been explored. For example, [1] also constructed a dataset using similar constraints for instruction fine-tuning.
2. The dataset may not be very useful. Specifically, because the rules are too vague/arbitrary, the extracted ground truth is not useful for the evaluation process: the authors only use them for comparing fluency. In addition, since the rules can be arbitrarily designed, this compiled dataset does not hold much value, because many similar datasets can be compiled with different engineering details.

[1] Controlled text generation with natural language instructions. https://proceedings.mlr.press/v202/zhou23g/zhou23g.pdf

**Questions:**

In the limitations, the authors mentioned potential problems with filtering and processing functions. Could the authors elaborate on what issues might exist for the extraction process?

---

> ### Author Response · Authors · 2023-11-16
> **Thanks**
>
> Thanks for your feedback! Let us know if your have followup questions.
>
> 1. **The idea is not very novel, and similar ideas have already been explored. For example, [1] also constructed a dataset using similar constraints for instruction fine-tuning.**
>
>     * Thanks for bringing up this concurrent work, which we have cited and discussed in the revised paper!
>     * It focuses on turning constraints into NL instructions and fine-tuning smaller language models such as T5-11B, and the constraints themselves are fairly simple and limited (only 5 types). In contrast, Collie focuses on making constrained text generation more flexible, customizable, and harder, challenging even SoTA LLMs like GPT-4.
>     * We believe these two efforts are orthogonal and can be complementary --- we can use these harder and more flexible constraints from Collie-v1 and other Collie constructed datasets to diagnose and analyze LMs (as shown in the paper), and improve the instruction tuning of LMs.
>
> 2. **The dataset may not be very useful. Specifically, because the rules are too vague/arbitrary, the extracted ground truth is not useful for the evaluation process: the authors only use them for comparing fluency. In addition, since the rules can be arbitrarily designed, this compiled dataset does not hold much value, because many similar datasets can be compiled with different engineering details.**
>
>     * The extracted ground truth text is not only useful for comparing fluency, but also to ensure the extracted constraint values are solvable using natural text. We believe it is actually  important to evaluate constraint satisfaction using the constraints instead of the extracted text, as there could be very diverse generations that all satisfy the same constraints.
>     * As stated in the paper, our main contribution is twofold: the Collie framework for constructing constrained text generation tasks, and the Collie-v1 dataset using a concrete and preliminary set of constraint types. The facts that “rules can be arbitrarily designed” and “many similar datasets can be compiled with different engineering details” are exactly the merit of our proposed Collie framework --- while previous datasets feature a fixed set of simple constraints and ad-hoc pipelines for data collection, the Collie framework allows automatic and scalable construction of arbitrary grammar-expressible constraints.
>
> 3. **In the limitations, the authors mentioned potential problems with filtering and processing functions. Could the authors elaborate on what issues might exist for the extraction process?**
>     * Every text corpus has artifacts (e.g., tables, footnotes, references, headings that might not be natural sentences or paragraphs). We design filtering and processing functions to patch or eliminate passages with these artifacts. However, it is impossible to design a filtering system that is perfect, so there could be downstream reference texts or constraints that look unnatural. This is more of a fundamental limitation with any approach that leverages text in the wild, and not a limitation specific to Collie or CTG construction. We have revised the limitation part to better explain this, thanks!

---

> > ### Comment · Reviewer_PuAQ · 2023-11-20
> >
> > Thanks for the detailed response!
> > Although I do not entirely agree, I am willing to raise my score from 5 to 6.

---

### Official Review · Reviewer_nWAg · 2023-11-01

**Soundness:** 4 excellent
**Presentation:** 4 excellent
**Contribution:** 4 excellent
**Rating:** 10
**Confidence:** 4

**Summary:**

This paper introduces a grammar based framework called COLLIE for constrained text generation at varying levels of specifications. Additionally it proposes a development tool for automated extraction of task instances given a constraint structure and text corpus. Furthermore, the paper constructs a dataset from three sources - Wikipedia, CCNews, and Project Gutenberg using the previously mentioned framework and calls it COLLIE-v1. The resultant dataset COLLIE-v1 is constructed using manually crafted constraints and is used to analyze LLM performances as well as highlight their shortcomings. The paper focuses on five of the most prominent LLMs, namely GPT-3.5,GPT-4,PaLM, Vicuna-7B and Alpaca-7B.

Interestingly, the COLLIE framework enables flexible, extensible and dynamic constraint construction that can co-evolve with the upcoming LLMs and help in understanding their shortcomings to better solve them.

COLLIE can thus help in not only evaluating and benchmarking LLMs but also help in constraint text generation independently.

**Strengths:**

1. Well written paper with evaluation on competitive LLM baselines.

2. Combination of rule based and neural based generation enables NLP grounded generations

3. Open-sourcing of code and the related dataset for promoting further research.

4. Comprehensive analysis to highlight the shortcoming of current LLMs that needs to be addressed.

**Weaknesses:**

1. Some important details for instruction rendering should be moved to the main paper.

2. The paper mentions that the technique can be used for constraining words, word blacklisting, however a qualitative analysis is missing for the same in the current version.

**Questions:**

Section 5: Performance enhancement through feedback - It might help to list down the details of how the feedback is being used.

---

> ### Author Response · Authors · 2023-11-16
> **Thanks**
>
> Thanks for your positive comments! Let us know if you have further feedback.
>
> 1. **Some important details for instruction rendering should be moved to the main paper.**
>
>     * As per your suggestion, we have moved more details for instruction rendering to the main paper (section 3, in blue).
>
> 2. **The paper mentions that the technique can be used for constraining words, word blacklisting, however a qualitative analysis is missing for the same in the current version.**
>
>     *  By word blacklisting we mean something similar to constraint para02: "Generate a paragraph with at least 4 sentences, but do not use the words “the”, “and” or “of”." We have complete model outputs in our supplementary material, but we can also add some of these examples to the appendix. Thank you for the suggestion.
>
> 3. **Section 5: Performance enhancement through feedback - It might help to list down the details of how the feedback is being used.**
>
>     * Thanks! We have moved some details for use of feedback to the main paper, as per your suggestion (section 5.1, in blue).

---

### Author Response · Authors · 2023-11-16
**General Response**

We thank all reviewers for their time and feedback! **We have revised the draft to incorporate all comments (in blue)**.

To briefly re-iterate the main contributions of our paper, we propose both the Collie framework and the Collie-v1 dataset.
* Collie-v1 dataset is much more systematic, diverse, and harder than existing CTG datasets. It serves to challenge even start-of-the-art LLMs like GPT-4 (while existing datasets cannot), and reveal useful insights about LLMs (see Section 5.1, e.g., position effect).
* The Collie framework allows easy and modular changes of the grammar, constraint types, used corpora, new evaluation metrics or instruction rendering rules, etc. to automatically construct new CTG datasets that could incorporate more concepts (e.g., POS tag, shown in Appendix B.5) or functional constraints. We release all code and data to support future research and development.

---

### Meta-Review · Area_Chair_SrVX · 2023-12-06

**Metareview:**

The paper addresses data synthesis for constrained text generation, where the authors combine both rule-based systems and LLMs. Specifically, the authors design a set of compositional grammar rules for specifying the constraints. Then, LLMs are prompted to generate candidate sentences, which are then verified by a symbolic checker. Results show that the synthetic data may challenge state-of-the-art LLMs.

Reviewers are generally interested in the methodology.

**Justification For Why Not Higher Score:**

The proposed methods are interesting.

**Justification For Why Not Lower Score:**

The goal of this paper is to generate synthetic data. Although it can challenge LLMs in certain scenarios, it's questionable whether the synthetic data can replace real ones (for example, it's reported that if LLM is continually finetuned on synthetic data, it may degenerate).

---

### Decision · Program_Chairs · 2024-01-16

Accept (poster)